# A 30-year reconstruction of the Atlantic meridional overturning circulation shows no decline

Emma L. Worthington[1], Ben I. Moat[2], David A. Smeed[2], Jennifer V. Mecking[2], Robert Marsh[1], and
Gerard D. McCarthy[3]

[1]University of Southampton, European Way, Southampton, SO14 3ZH, UK
[2]National Oceanography Centre, European Way, Southampton, SO14 3ZH, UK
[3]ICARUS, Department of Geography, Maynooth University, Maynooth, Co. Kildare, Ireland

**Correspondence:** Emma L. Worthington (emma.worthington@soton.ac.uk)

**Abstract.** A decline in Atlantic meridional overturning circulation (AMOC) strength has been observed between 2004 and 2012
by the RAPID array with this weakened state of the AMOC persisting until 2017. Climate model and paleo-oceanographic re-
search suggests that the AMOC may have been declining for decades or even centuries before this, however direct observations
are sparse prior to 2004, giving only 'snapshots' of the overturning circulation. Previous studies have used linear models based
on upper layer temperature anomalies to extend AMOC estimates back in time, however these ignore changes in the deep
circulation that are beginning to emerge in the observations of AMOC decline. Here we develop a higher fidelity empirical
model of AMOC variability based on RAPID data, and associated physically with changes in thickness of the persistent upper,
intermediate and deep water masses at 26°N and associated transports. We applied historical hydrographic data to the empir-
ical model to create an AMOC time series extending from 1981 to 2016. Increasing the resolution of the observed AMOC to
approximately annual shows multi-annual variability in agreement with RAPID observations, and that the downturn between
2008 and 2012 was the weakest AMOC since the mid-1980s. However, the time series shows no overall AMOC decline as
indicated by other proxies and high resolution climate models. Our results reinforce that adequately capturing changes to the
deep circulation is key to detecting any anthropogenic climate change-related AMOC decline.

*Copyright statement.* TEXT

## 1 Introduction

In the northern hemisphere, the Atlantic meridional overturning circulation (AMOC) carries as much as 90% of all the heat
transported poleward by the subtropical Atlantic Ocean (Johns et al., 2011), with the associated release of heat to the overlying
air helping to maintain north-western Europe's relatively mild climate for its latitude. The AMOC also transports freshwater
towards the equator, and the associated deep water formation moves carbon and heat into the deep ocean (Kostov et al., 2014;
Winton et al., 2013; McDonagh et al., 2015). A significant change in AMOC circulation is thus likely to have an impact on
the climate of north-western Europe and further afield, with possible influences on global hydrological and carbon cycles.

Although the Intergovernmental Panel on Climate Change (IPCC) says that it is unlikely that the AMOC will stop this century, they state with medium confidence that a slowdown by 2050 due to anthropogenic climate change is very likely (Stocker et al., 2013).

The importance of the AMOC means that since 2004 it has been observed by the RAPID mooring array at 26°N. The resulting observations have highlighted the great variability in AMOC transport on a range of timescales (Kanzow et al., 2010; Cunningham et al., 2007), including a decline in AMOC strength between 2004 and 2012 (Smeed et al., 2014). This reduced state persisted in 2017 (Smeed et al., 2018). The decrease is more likely to be internal variability rather than a long-term decline in response to anthropogenic forcing (Roberts et al., 2014), which the time series is currently too short to detect. Although the

AMOC has been well-observed at 26°N since 2004, prior to this estimates of AMOC strength were restricted to instances of transatlantic hydrographic sections along 24.5°N in 1957, 1981, 1992, 1998 and 2004, which provided only snapshots of the overturning circulation strength (Bryden et al., 2005). There are extensive additional hydrographic data around 26°N, particularly at the western boundary, but these are insufficient to reconstruct the AMOC conventionally (Longworth et al., 2011). Due to the limited availability of hydrographic data, proxies have been used to reconstruct the AMOC time series

earlier than 2004.

In one proxy reconstruction, Frajka-Williams (2015) used sea-surface height from satellite altimetry to estimate trans-basin baroclinic transport at 26°N between 1993 to 2014. In another, Longworth et al. (2011) used temperature anomaly at the western boundary as a proxy for geostrophic transport within the upper 800 m, or thermocline layer, finding the temperature anomaly at 400 dbar explained 53% of the variance in thermocline transport. However, both Longworth et al. (2011) and Frajka-Williams

(2015) used single layer models that do not account for the variable depth structure of the AMOC in the subtropics.

At 26°N, the dynamics of the AMOC involve multiple water masses flowing in opposite directions in different layers, driven by the changing density structure with depth (Figure 1a). Within the permanent thermocline layer, which reaches as deep as 800 m on the western boundary and 600 m on the eastern, isopycnals rise towards the eastern boundary, indicative of southward flow (Hernández-Guerra et al., 2014). Below the thermocline, isopycnals deepen towards the east, and the resulting transport profile

(Figure 1c) shows a small northwards transport centered around 1000 m sandwiched between southwards transports above and below. Although referred to by RAPID as Antarctic Intermediate Water (AAIW), both AAIW and Mediterranean Water are observed between 700-1600 m on the eastern boundary, with the relative contribution of each varying seasonally (Fraile-Nuez et al., 2010; Machín and Pelegrí, 2009; Hernández-Guerra et al., 2003). The transport profile also shows North Atlantic Deep Water (NADW), which has two distinct layers: Upper (UNADW) above 3000 m, primarily formed in the Labrador Sea (Talley

and McCartney, 1982); and Lower (LNADW) below 3000 m, which has its origins in the overflows from the Nordic Seas (Pickart et al., 2003). Changes observed in one NADW layer are not necessarily observed in another. (Smeed et al., 2014) found the reduction in AMOC strength between 2004 and 2012 was seen in LNADW but not UNADW, while Bryden et al. (2005) found LNADW transport estimated from transatlantic hydrographic sections at 25°N decreased from -15 Sv in 1957 to less than -7 Sv in 1998 and 2004 but the UNADW transport remained between -9 and -12 Sv. Below the NADW layers,

there is a small northwards transport below 5000 m, Antarctic Bottom Water (AABW), that flows along the western side of the Mid-Atlantic Ridge. The partition between the upper southwards and deep southwards transports defines the strength of

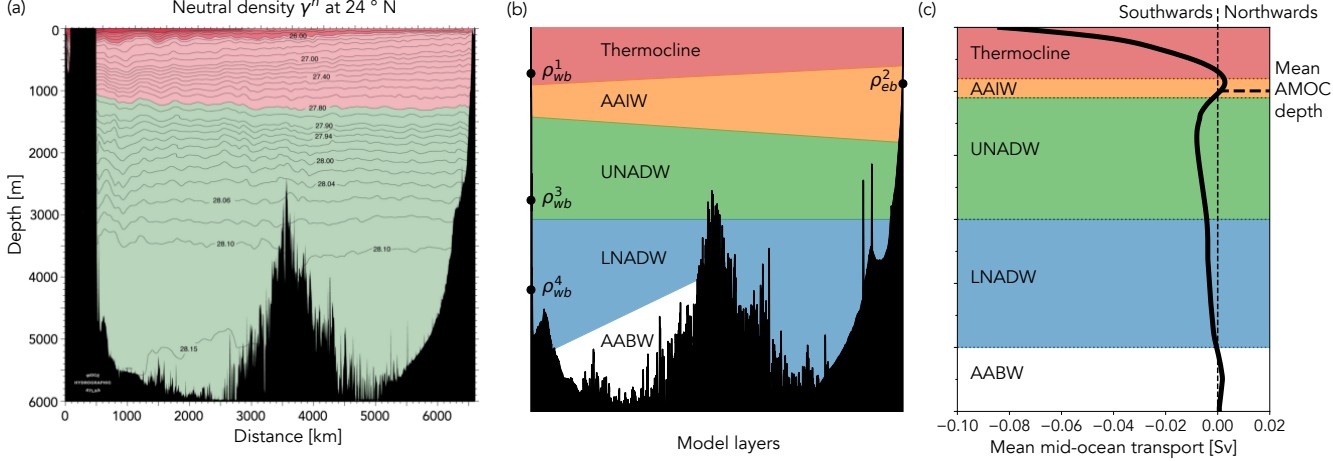

**Figure 1.** (a) World Ocean Circulation Experiment (WOCE) North Atlantic A05 section of neutral density $\gamma^n [kg\,m^{-3}]$ at 24°N. From the WOCE Atlantic Ocean Atlas Vol. 3. (K. Koltermann, K.P., Gouretski, V.V., Jancke, 2011) (b) Schematic of four dynamic layers to be represented within the regression model by density anomalies at the western and eastern boundaries at a depth within each layer. The density anomalies are represented by the circular markers. (c) Profile of RAPID-estimated mean mid-ocean transport and the resulting northwards and southwards layer transports. Mean AMOC depth is around 1100 m.

the overturning circulation: a weak AMOC is associated with a greater recirculation within the upper layers of the thermocline and weaker deep return flow; a stronger AMOC is associated with weaker thermocline recirculation and stronger deep NADW transport. For an empirical model to more fully represent AMOC dynamics, in particular lower frequency changes, we suggest
that it must represent these deeper layers. A layered model interpretation of the density structure and the associated water mass transports is shown in Figure 1b.

Here, we revisit the approach of Longworth et al. (2011) by using linear regression models to represent the AMOC, and develop the method further to include additional layers representative of the deep circulation. Section 2 describes how we trained and validated our statistical model using the RAPID dataset, and how we selected historical hydrographic data to apply
to the model. Section 3 describes how these hydrographic data were used to create an extended timeseries of AMOC strength from 1982 to 2016. In Sections 4 and 5, we discuss the implications of creating the longest observational timeseries of AMOC strength that incorporates variability in the deep NADW layers, and acknowledge the limitations of using an empirical model.

## 2   Methods

### 2.1   Model data

Our regression models were trained on RAPID data from 27 May 2006 to 21 February 2017 (Smeed et al., 2017). RAPID data are available from 7 April 2004, but we used only data obtained after the collapse of the main western mooring, WB2,

between 7 November 2005 and 26 May 2006. McCarthy et al. (2015) describes in detail how RAPID measures the AMOC, but it is described briefly here as we use both its results and the interim data created during the calculation. AMOC transport at 26°N ($T_{amoc}$) is estimated by combining four directly observed components (Equation 1): Gulf Stream transport within the Florida Straits ($T_{flo}$), which is measured by submarine cables and calibrated by regular hydrographic sections (Baringer and Larsen, 2001; Meinen et al., 2010); Ekman transport ($T_{ek}$), which here is calculated from ERA-Interim reanalysis wind fields; Western Boundary Wedge transport ($T_{wbw}$), which is obtained from direct current measurements over the continental slope between the Bahamas and the WB2 mooring at 76.75°W; and the internal transport ($T_{int}$), the basin-wide geostrophic transport calculated from dynamic height profiles described below, relative to a reference depth of no motion at 4820 dbar. This reference depth is selected as the approximate depth of the interface between the southwards LNADW and the deeper northwards AABW (McCarthy et al., 2015). To the sum of these four components, a depth-dependent external transport ($T_{ext}$) is added to ensure mass is conserved and that there is zero net flow across the section. The assumption of zero net flow holds on timescales longer than 10 days (Kanzow et al., 2007; Bryden et al., 2009).

$$T_{amoc}(t, z) = T_{flo}(t, z) + T_{ek}(t, z) + T_{wbw}(t, z) + T_{int}(t, z) + T_{ext}(t, z) \tag{1}$$

The internal geostrophic transport, $T_{int}$, is estimated from dynamic height profiles. These are created by merging data from individual moorings to create four profiles for each of the western and eastern boundaries and the western and eastern sides of the Mid-Atlantic Ridge. For example, at the western boundary, most data comes from instruments deployed on the WB2 mooring, but additional data from the deeper, more eastern WBH2 and WB3 moorings are used to cover the full depth. This results in vertical profiles of temperature and salinity with sparse resolution, which are then vertically interpolated using a monthly climatology for each location. As RAPID data are vertically interpolated over a pressure grid, depth will henceforth be reported in decibars (dbar) rather than metres. These four merged and interpolated temperature and salinity profiles are used to calculate dynamic height (referenced to 4820 dbar), which is then extrapolated to the surface using a seasonal climatology.

The strength of the AMOC is then the maximum of $T_{amoc}$ integrated over depth from the surface, i.e., the maximum of the transport streamfunction; and the AMOC depth ($z_{amoc}$) is the depth of that maximum at each time step, usually around 1100 m. The upper mid-ocean (UMO) transport ($T_{umo}$) is defined as the mid-ocean transport ($T_{mo}$) integrated between the surface and the AMOC depth (Equation 2), where the mid-ocean transport is the sum of the internal, external and Western Boundary Wedge transports. This net southwards UMO transport includes the southward gyre recirculation and the northwards Antilles Current.

$$T_{umo}(t) = \int^{z_{amoc}} T_{mo}(t, z)\, dz = \int^{z_{amoc}} [T_{int}(t, z) + T_{wbw}(t, z) + T_{ext}(t, z)]\, dz \tag{2}$$

## 2.2 Developing the model

For use in the regression models, absolute salinity, conservative temperature and in situ density were calculated from the gridded in situ temperature and practical salinity data created during the RAPID calculations. As all AMOC transports are

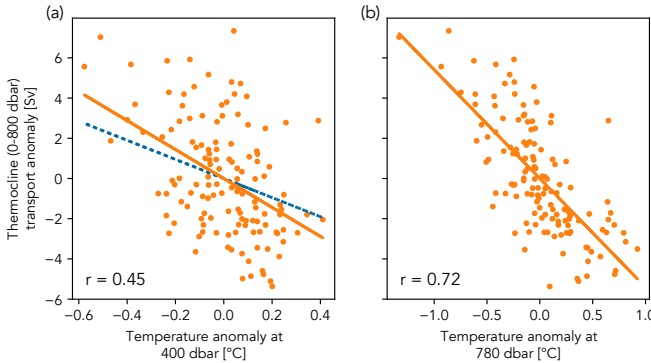

**Figure 2.** Linear regression of the monthly mean RAPID thermocline (0-800 m) transport anomaly on the monthly mean conservative temperature anomaly at (a) 400 dbar and (b) 780 dbar from the RAPID western boundary profiles. The orange line is the regression equation for these data, the blue dashed line in (a) shows the equivalent regression from Longworth et al. (2011) for the same data. The Pearson's correlation coefficient (r) is shown for each model regression.

filtered during the RAPID calculation, the same Butterworth 10-day, low-pass filter was also applied to the salinity, temperature and density data; the filtered data were then averaged from a 12-hourly resolution to a monthly mean. Anomalies of these data
and RAPID-estimated transports were created by subtracting the mean between 27 May 2006 and 21 February 2017, and these monthly mean anomalies were used to train all our regression models.

We revisited the linear regression made by Longworth et al. (2011), an ordinary least-squares (OLS) regression between the conservative temperature anomaly at 400 dbar and the thermocline transport anomaly. Longworth et al. (2011) suggested that increasing southwards thermocline transport (a negative transport anomaly) causes more warm water to recirculate close to the
western boundary, and so the temperature at any particular depth within the thermocline will increase (a positive temperature anomaly). They chose 400 dbar as both the mid-point of the thermocline layer, and a depth at which every profile had a sensor deployed within 50 m. We used 132 monthly mean conservative temperature anomalies at 400 dbar from the RAPID western boundary profile, compared to their 39 historic CTD profiles. Our regression however, shown in the scatter plot in (Figure 2a), explained only 20% of the variance of the thermocline transport anomaly, compared with 53% for the original result based on
the shorter time series. To investigate whether the regression fit could be improved by using the temperature anomaly from a different depth, we used an algorithm to repeat the same regression using the conservative temperature anomaly every 20 dbar from 220 to 800 dbar, and report at which depth the highest explained variance, or adjusted $R^2$ value, was found. The highest explained variance found by the algorithm was 51% for the regression using the western boundary temperature anomaly at 780 dbar (Figure 2b. Changing the regression to use the density anomaly made very little difference, increasing the adjusted $R^2$ to
0.54, with the anomaly again at 780 dbar.

We expanded this one layer model by creating multiple linear regression models using two, three and four explanatory variables to represent two, three and four layers respectively, reflective of the water mass and circulation depth structure. We used RAPID-defined layers: the thermocline between the surface and 800 dbar; Antarctic Intermediate Water (AAIW) between

800 and 1100 dbar; Upper North Atlantic Deep Water (UNADW) between 1100 and 3000 dbar; and Lower North Atlantic Deep Water (LNADW) between 3000 and 4820 dbar (Figure 1b). UMO transport ($T_{umo}$), as the main contributor to the AMOC, was preferred to thermocline transport as the dependent variable for all subsequent regression models. Variability of the AMOC is dominated by the western boundary (Elipot et al., 2014; Bryden et al., 2009); and Frajka-Williams et al. (2016) showed strong positive correlation between UMO transports and isopycnal displacements on the western boundary around 820 m, and negative correlation between LNADW transport and isopycnal displacements on the western boundary between 1500 m and the bottom. Western boundary density anomalies were thus chosen as the independent variables representing each layer, with the exception of AAIW. The seasonal cycle of the AMOC is driven largely by seasonality at the eastern boundary (Chidichimo et al., 2010; Pérez-Hernández et al., 2015). The annual maximum northwards transport at the eastern boundary and the AMOC occur around October (Vélez-Belchí et al., 2017), and is driven by changes in the circulation of the Canary Current (Casanova-Masjoan et al., 2020; Hernández-Guerra et al., 2017), and at intermediate depths (700-1400 dbar) by seasonal changes in the Intermediate Poleward Undercurrent (Hernández-Guerra et al., 2017; Vélez-Belchí et al., 2017). Eastern boundary density anomalies have maximum sub-surface variability around 1000 dbar (Chidichimo et al., 2010), so the AAIW layer was represented by an eastern boundary density anomaly between 800 and 1100 dbar. The multiple linear regression equation (Equation 3) shows the four explanatory variables — three western and one eastern boundary density anomalies — representing the thermocline ($\rho_{wb}^{z1}$), AAIW ($\rho_{eb}^{z2}$), UNADW ($\rho_{wb}^{z3}$), and LNADW ($\rho_{wb}^{z4}$) layers. The superscripts z1, z2, z3, and z4 indicate the anomaly depths to be identified by algorithm.

$$T_{umo}\left(t\right) = \alpha\ \rho_{wb}^{z1}\left(t\right) + \beta\ \rho_{eb}^{z2}\left(t\right) + \gamma\ \rho_{wb}^{z3}\left(t\right) + \zeta\ \rho_{wb}^{z4}\left(t\right) \tag{3}$$

Initially a model with two explanatory variables representing the thermocline ($\rho_{wb}^{z1}$) and AAIW ($\rho_{eb}^{z2}$) layers was implemented, then variables representing the UNADW ($\rho_{wb}^{z3}$), and finally the LNADW ($\rho_{wb}^{z4}$) were added. All possible combinations of density anomaly depths within each layer were used to run the regression to find which combination gave the maximum adjusted $R^2$ value. For example, for the two-layer model, each western boundary density anomaly every 20 dbar between 220 and 780 dbar was combined in turn with each eastern boundary density anomaly every 20 dbar between 800 and 1080 dbar, and regressed against the UMO transport anomaly. Due to the number of iterations required by adding the fourth variable, the western boundary density anomalies representing the UNADW and LNADW in this model were chosen every 100 dbar rather than every 20.

The OLS regressions were checked against a number of assumptions that should hold for a linear regression model to be fit for purpose, for example, a known issue for those models based on time series is auto-correlation of residuals. We found that all our OLS models, whether using simple or multiple linear regression, showed autocorrelation of residuals, indicated by Durbin-Watson values between 0 and 1. An alternate linear regression model was used, the generalized least-squares model with auto-correlated errors (GLSAR), which models autocorrelation of residuals for a given lag (McKinney et al., 2019). For each of our models, autocorrelation was significant for a lag of one month, which is consistent with Smeed et al. (2014), who show the decorrelation length scale associated with the AMOC is 40 days.

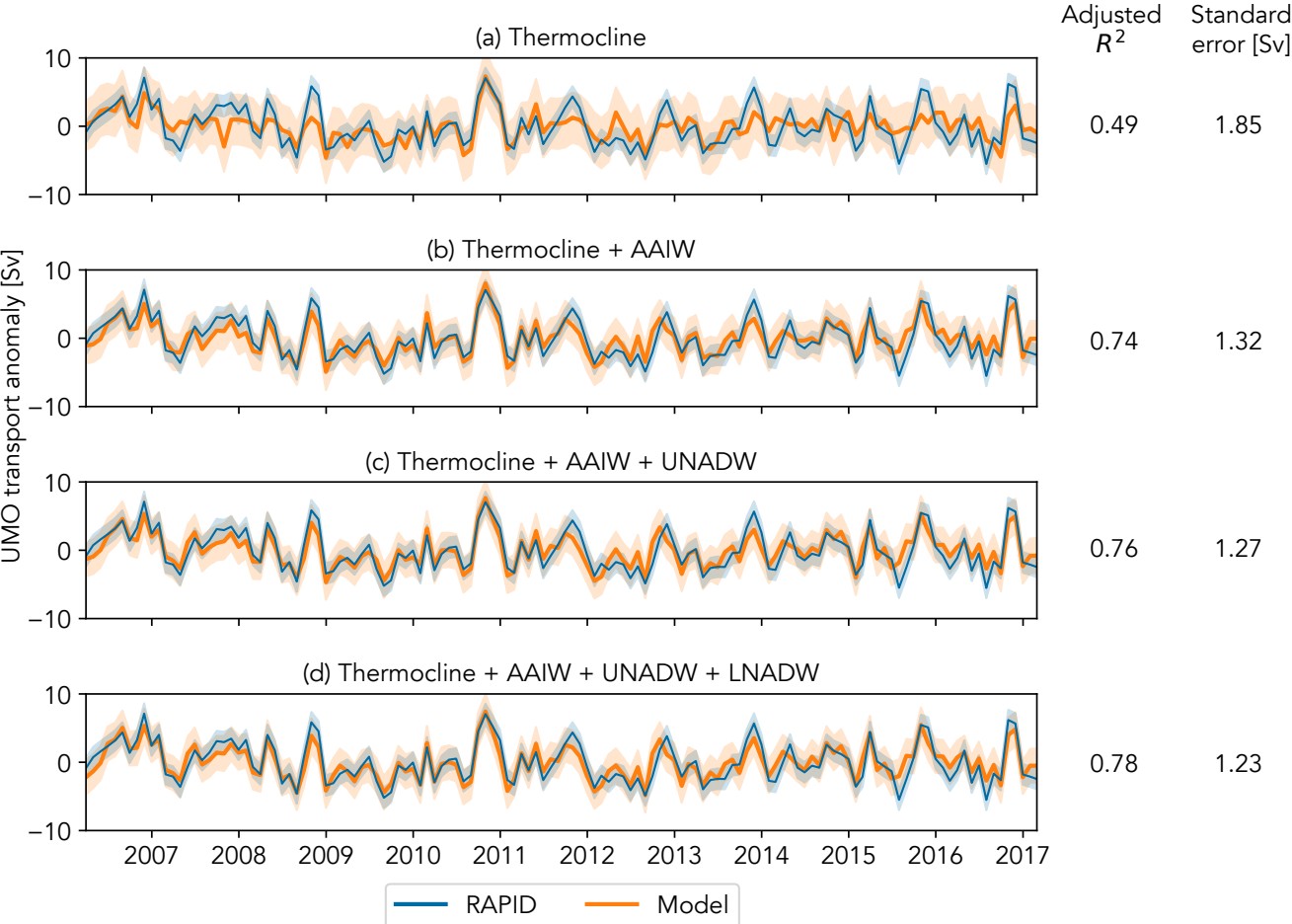

**Figure 3.** Comparison of UMO transport anomaly predicted by GLSAR(1) regression models using one to four layers with the UMO transport anomaly observed by RAPID. The layers represented by the regression independent variables are shown above each plot, and the model $R^2$ value, adjusted for the degrees of freedom of each model; and standard error of the regression, are shown to the right. The orange shading around each model prediction line shows the 95% prediction interval. The blue shading around the observed transport shows the 1.5 Sv uncertainty estimated by McCarthy et al. (2015)

## 2.3 Evaluating the model

The simplest model selected by algorithm, regressing UMO transport anomaly on the western boundary density anomaly at 780 dbar, gives an adjusted $R^2$ value of only 0.49, shown in the top time series plot (Figure 3a). This plot also shows relatively large model prediction intervals (orange shading), which give the range of UMO transport anomalies that we have 95% confidence will occur for that combination of boundary density anomalies. However, adding the eastern boundary density anomaly at 1020 dbar ($\rho_{eb}^{1020}$) increases the maximum adjusted $R^2$ to 0.74 (Figure 3b) and reduces the prediction intervals. Adding the

160

$\rho_{wb}^{z3}$ and then the $\rho_{wb}^{z4}$ western boundary density anomalies further increases the adjusted R$^2$ value of the model to 0.76 and 0.78 respectively (Figure 3c and d). Although increasing layers from two to three to four does not increase the adjusted R$^2$ greatly, it does reduce the standard error of the regression, from 1.85 Sv for the single layer model, to 1.32, 1.27 and 1.23 Sv for the two, three and four-layer models. The algorithm also selects slightly different density anomaly depths for these last two regressions: 720, 980 and 1200 dbar when three variables are used; and 740, 980, 1200 and 3000 dbar when all four are included. As using explanatory variables to represent all four layers gives a regression model that explains the greatest variance in the UMO transport anomaly and has the lowest standard error, it was selected to apply to the historical hydrographic data. The resulting multiple linear regression equation (Equation 4) shows the depths chosen for each of the four density anomalies by the algorithm, and the coefficients for each.

$$T_{umo} = 40.5\ \rho_{wb}^{740} - 98.6\ \rho_{eb}^{980} + 46.7\ \rho_{wb}^{1200} + 46.8\ \rho_{wb}^{3000}, \tag{4}$$

The selected model was cross-validated using a 30/70% training/testing split of the RAPID data, to investigate its suitability for predicting over a period much longer than the almost 11 years of RAPID data used to train the full model. The two cross-validation models, trained using the first and last 30% of the RAPID data, both predict UMO transport anomalies for the remaining 70% that agree well with the observations (r=0.88). The full model was then tested against new RAPID data not used to train it, made available following the most recent expedition and covering the period from February 2017 to November 2018 (Smeed et al., 2019). The model-predicted UMO transport anomaly shown in Figure 4a shows that it reproduces the trends and variability, although not always the magnitude, of the observed values well (r=0.75). It also shows that when the eastern boundary density anomaly at 980 dbar is replaced with a monthly climatology, the trends and variability are also well captured (r=0.71). The reason that a climatology was tested will be discussed in subsection 2.5 when we describe the selection of historical hydrographic data.

Our model was trained on monthly mean density anomalies, but was to be used with hydrographic data from much shorter periods of a day or two. To evaluate how well these 'snapshot' profiles represented the longer periods, we simulated them by randomly selected 20 single points from the 7961 available 12-hourly values from the most recent RAPID data. These were applied to the model and the predicted UMO compared to the observed monthly mean UMO for the same time, with the model error being the difference between the two. Bootstrapping the model prediction showed that around 65% of the observed UMO values were within the prediction interval of the corresponding model UMO. The standard deviation of the bootstrapped model errors was 2.8 Sv.

The CTD profiles to be applied to the regression model to estimate UMO transport occur at irregular intervals, so to allow comparisons between periods, we calculated the mean transport anomalies for a given time window. Additionally, we calculated the weighted rolling mean for the transport anomalies with the RAPID annual cycle removed, using a Gaussian distribution over the same time window. Since Smeed et al. (2014) calculated means for two 4-year periods, 2004–2008 and 2008–2012, from April to March, we used the same four-year window for both the period mean and weighted rolling mean to allow a direct comparison. Since the UMO is a transport specific to RAPID, we also estimated the AMOC by adding the model-derived

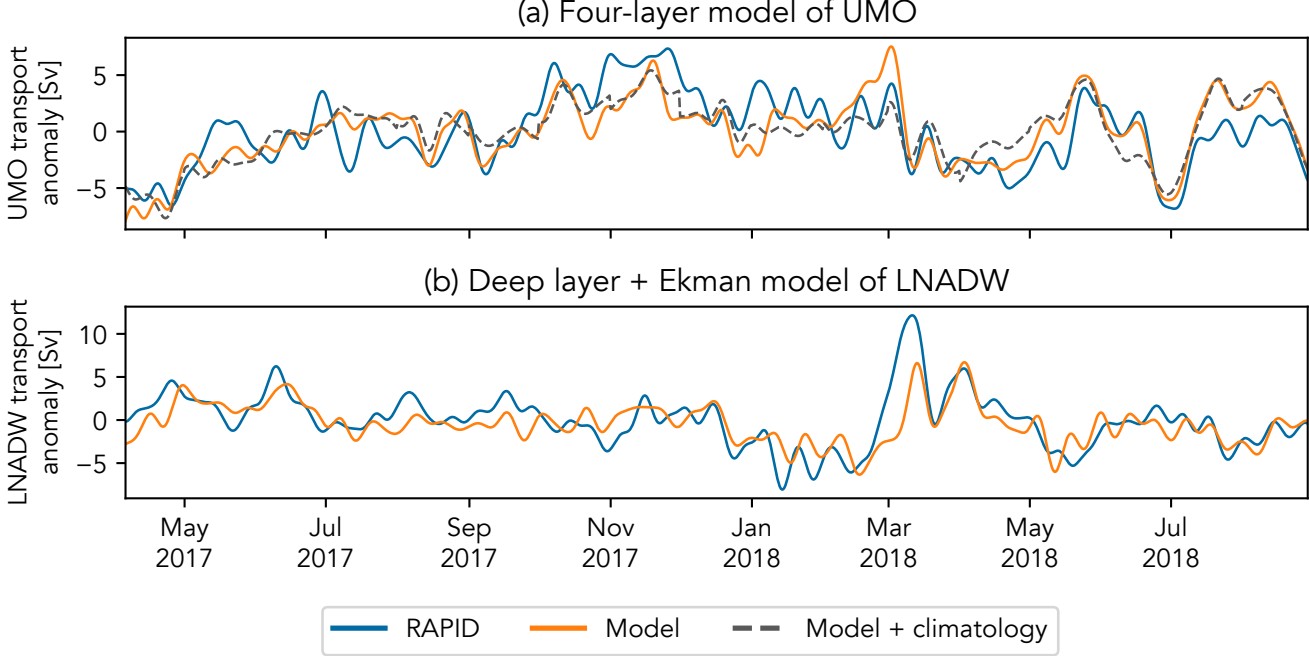

**Figure 4.** (a) UMO transport anomaly estimated from the most recent RAPID observations (blue) compared against the UMO transport anomaly predicted by the four-layer regression model (orange) using density anomalies derived from the same RAPID data. The RAPID data were 12-hourly and 10-day filtered. The grey line shows the model prediction where the eastern boundary density anomaly at 980 dbar is replaced by a monthly climatology. (b) LNADW transport anomaly estimated from the most recent RAPID observations (blue) compared against the LNADW transport anomaly predicted by the regression model combining the western boundary density anomaly at 3040 dbar (orange) derived from the same RAPID data and the Ekman transport.

UMO transport to the monthly mean Florida Current and Ekman transport anomalies for the same date. The Florida Current data were the Western Boundary Time Series daily mean transport estimates from submarine cable voltage, and the Ekman data were the same ERA-interim reanalysis-derived Ekman transports as used by RAPID. The Florida Current data have a gap between 22 October 1998 and 19 June 2000. There were no selected hydrographic profiles during this gap, but this period was 200 filled with the time series mean to allow the four-year mean and weighted rolling mean to be calculated for the Florida Current transport. To give overall transports, the relevant mean transports from the RAPID data used to train the model (27 May 2006 to 21 February 2017) were added to the four-year and rolling mean anomalies.

To evaluate the co-variability of the density anomaly selected to represent each water mass transport and the observed UMO transport anomaly, we determined the coherence between them using a multi-taper spectrum following Percival and Walden 205 (1998). This method reduces spectral leakage while minimising the data loss associated with other tapers. The number of tapers used was $K = 2p - 1$, where $p = 4$, and the 95% confidence level given by $1 - 0.05^{1/(K-1)}$. The western boundary density anomaly at 740 dbar, which represents the thermocline water mass transport, shows shows significant coherence with

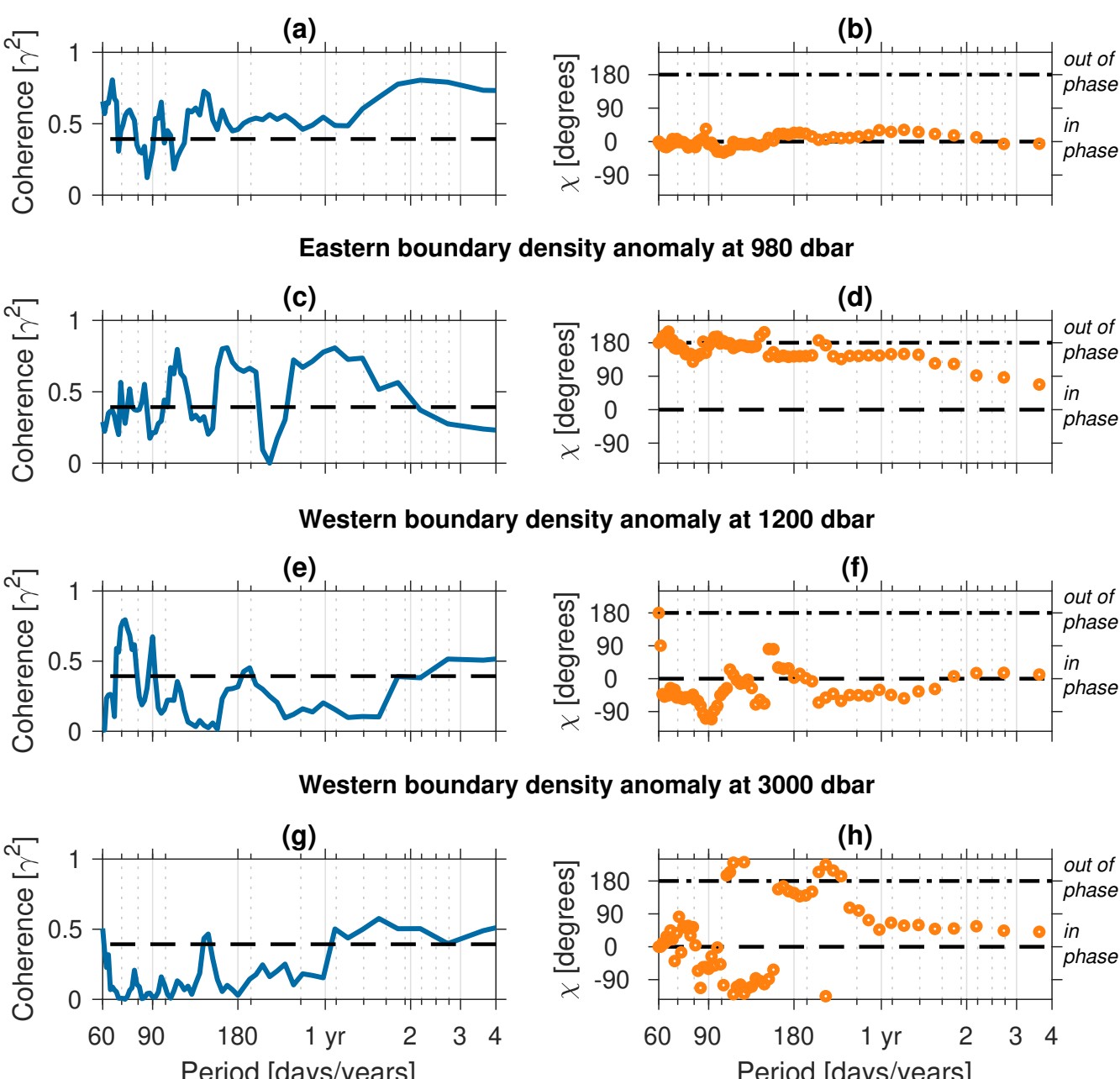

**Figure 5.** Multi-taper spectrum coherence (left) and phase relationship (right) between the UMO transport anomaly observed by RAPID and the density anomaly that is each independent variable in the linear regression. Significance of coherence (95% confidence) is indicated by the black dashed horizontal line in the left-hand figures. The horizontal black dashed lines in the right-hand figures show where the time series are exactly in phase, and exactly 180° out of phase.

the observed UMO transport at periods of around 65, 75 and 95 days; and then at all periods from 120 days to 4 years. The highest significance occurs for periods of around 65 days and 2–3 years (Figure 5a), and is in phase at all periods (Figure 5b). The eastern boundary density anomaly at 980 dbar, representing the AAIW water mass transport, shows significant coherence for periods between 100 and 120 days, 150 to 200 days, and between 280 days and 1.5 years, with the strongest coherence at around 160 days and just over 1 year (Figure 5c). The coherence for this variable is out of phase for most periods with significant coherence, (Figure 5d)), which is consistent with its negative coefficient. The western boundary density anomaly at 1200 dbar, representing the UNADW water mass transport, shows significant coherence for periods between 67 and 80 days, and around 90 days, and to a lesser extent around 200 days (Figure 5e). The significant coherence is approximately in phase, with the observed UMO transport anomaly lagging the 1200 dbar density anomaly slightly (Figure 5f). Finally, the western boundary density anomaly at 3000 dbar, representing the LNADW water mass transport, shows significant coherence only for periods at just under 140 days, and between just over 1 year, or 400 days (Figure 5g), and just over 2.5 years, or 990 days. For this latter period, the co-variability is also approximately in phase, with the 3000 dbar density anomaly lagging the UMO transport anomaly slightly (Figure 5h).

### 2.4 An additional model: reconstructing Lower North Atlantic Deep Water

The importance of the LNADW in the AMOC decline compared to the UNADW (Smeed et al., 2018) suggested an additional linear regression model between the LNADW transport anomaly and two independent variables; a western boundary density anomaly at a depth within the LNADW layer, and the Ekman transport anomaly. The Ekman transport is included in the model as Frajka-Williams et al. (2016) found that LNADW transport showed a deep baroclinic response to changes in Ekman transport. We applied a similar algorithm to the UMO regression model, repeating the regression for the deep density anomaly every 20 dbar between 3000 dbar and 4820 dbar, and reporting the maximum explained variance.

For the LNADW linear regression, the algorithm selected the western boundary density anomaly at 3040 dbar, close to the boundary between the upper and lower North Atlantic Deep Water layers at 3000 dbar. The resulting linear regression (Equation 5) has an adjusted $R^2$ value of 0.75, a standard error of 0.94 Sv, and the coefficients show that a positive anomaly in LNADW transport is associated with both a negative density anomaly and negative Ekman transport anomaly. This means a reduction in the deep southwards LNADW flow is linked to lower density water at 3040 dbar, which we would expect with a reduction in overturning. The inverse relation between LNADW and Ekman transports reflects the statistically significant inverse correlation (r = -0.58) found by Frajka-Williams et al. (2016) between the two transports.

$$T_{\text{lnadw}} = -175.9 \rho_{wb}^{3040} - 0.4\, T_{ekman} \tag{5}$$

This model was also tested using the RAPID data between February 2017 to November 2018, and the model-predicted LNADW transport anomaly shown in Figure 4b shows that it compares well with the RAPID-observed equivalent (r=0.73). The hydrographic profiles selected for use in the UMO empirical model were also applied to this LNADW model, and the four-year and weighted rolling means calculated using the same windows.

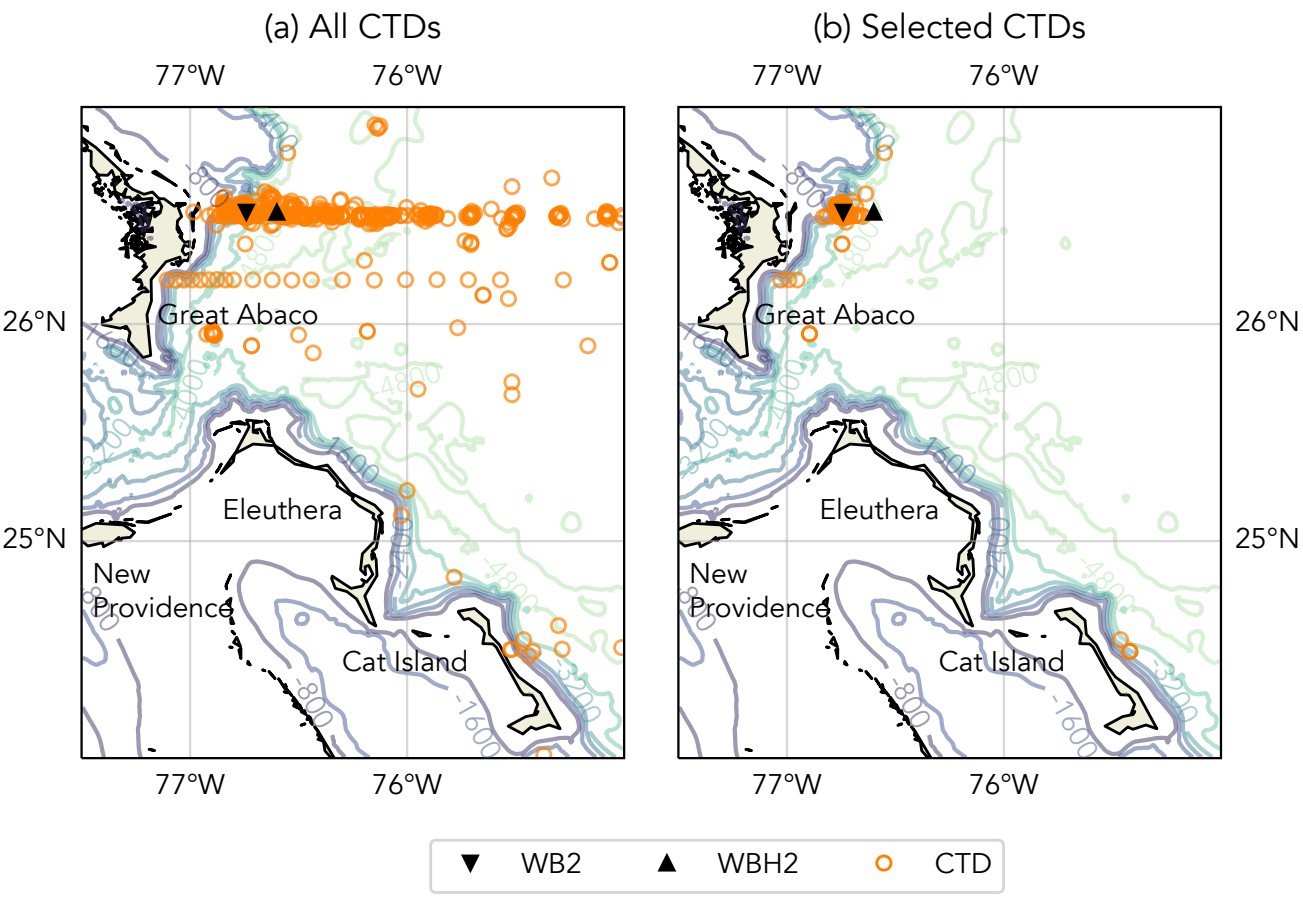

**Figure 6.** (a) All CTD profiles from the World Ocean Database 2018 in the region shown, compared with (b) those selected as being the most similar distance from the western boundary as the WB2 and WBH2 RAPID moorings.

## 2.5 Selecting historical hydrographic data

As Longworth et al. (2011) documented, between 1980 and 2017 there are many more hydrographic profiles close to the western boundary than the eastern. Since the eastern boundary density anomaly shows strong seasonal variability (Pérez-Hernández et al., 2015; Chidichimo et al., 2010), replacing it in the model with a monthly climatology from the RAPID eastern boundary data, as described in the previous subsection 2.3, allows us to use all available western boundary profiles, although at the cost of losing a little of the explained UMO transport variance.

Historical hydrographic data were obtained from the World Ocean Database 2018 (WOD2018) (Boyer et al., 2018), and from the datasets processed by Longworth et al. (2011), with duplicate data removed. Data were selected initially based on a region defined by latitude and longitude, from 24°N to 27°N, and from 75°W to 77.5°W. CTD profiles were then grouped by date, with a date group defined as separated by 3 days or more. From each group, we selected profiles based on similarity of

distance from the continental slope to the west as the RAPID WB2 and WBH2 moorings. We justify this as the AMOC shows meridional coherence of buoyancy anomalies within 100 km, and variability at the western boundary increases with distance from it (Kanzow et al., 2009). The RAPID western boundary profile that the model is based on uses the WB2 mooring for measurements down to around 3850 dbar, then the WBH2 and WB3 moorings below this. The distance from each mooring to the continental slope was calculated at the depths of the western boundary density anomalies that the algorithm selected. The same was done for each CTD profile. Then, for the profiles within each group, those with the most similar distance for each depth were selected. For example, for the density anomaly $\rho_{wb}^{740}$, the WB2 mooring is 13.8 km from the continental slope at 740 dbar. If there are 5 CTD profiles within the group, with distances from the slope at 740 dbar of 9.8 km, 12.4 km, 15.0 km, 18.9 km, and 27.5 km, then the profile that is 12.4 km away is selected. The same is done for the density anomalies $\rho_{wb}^{1200}$ and $\rho_{wb}^{3000}$. Thus between one and three CTD profiles are selected from each group to use in the model, and merged if required. The two regional plans in Figure 6, which compare all available CTD profiles and those selected for use in the model, show that the majority were located close to the RAPID western boundary mooring array.

## 3    Applying the model to historical hydrographic data

Initially, we used the western boundary density anomalies derived from the transatlantic sections at 24.5°N from 1981, 1992, 1998, 2004, 2010 and 2015 to estimate the UMO transport anomaly using the four-layer regression model. The error bars show the model prediction interval, which gives the range of UMO transport anomalies that we have 95% confidence will occur for that combination of boundary density anomalies. The uncertainty for the 1957 section model estimate was much larger than for the later section, and since no suitable hydrographic data before 1981 was available, it is omitted from our results. The sections from 1981, 1992, 1998 and 2004 were used by Bryden et al. (2005) to reconstruct the AMOC fully, and Figure 7 shows generally very good agreement between the model and Bryden et al. (2005) AMOC estimates. Only the 2004 estimate from Bryden et al. (2005) is not within the model's prediction interval, however the model 2004 estimate appears to be in better agreement with the RAPID observations. The model UMO estimates for 2010 and 2015 are also consistent with the RAPID UMO, reflecting the large downturn between late 2009 and early 2010 and a peak in 2015. The model consistently predicts a stronger AMOC than Bryden et al. (2005), with mean and maximum differences of 2.6 and 4.4 Sv.

When the model is applied to western boundary density anomalies from the selected hydrographic profiles, together with the eastern boundary climatology, the resolution of the UMO time series in Figure 8 is sufficient to show decadal and multi-annual variability. The standard deviation of the RAPID monthly mean anomalies is ±2.9 Sv, and the UMO transport anomaly estimated by the model is stronger southward than this in May 1985, March 1989, June 1993, February 2003, and March 2004, with each anomaly lower than -3.3 Sv. This is stronger than any negative anomaly predicted by the model during the RAPID period. The UMO is weaker southward than the RAPID standard deviation in September 1981, February 1986, July and August 1992, October 2004, and October and December 2015, with the anomaly again being greater than 3.5 Sv with the exception of July 1992. When compared directly to the RAPID observations, the model predictions generally agree well with

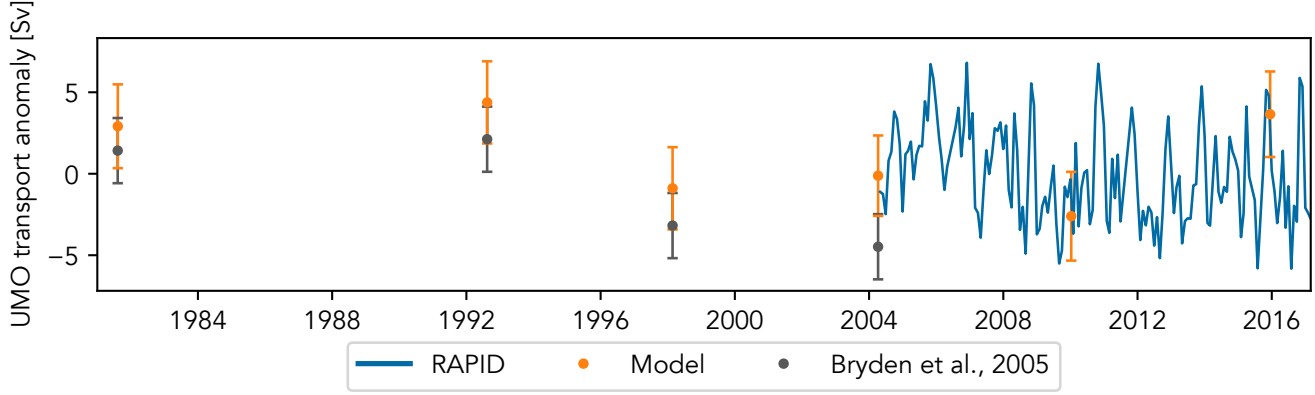

**Figure 7.** UMO transport anomaly estimated by the empirical model using density anomalies from six transatlantic hydrographic sections, compared to estimates from Bryden et al. (2005) and RAPID. The uncertainties shown for the model-derived values are the model's prediction intervals; the Bryden et al. (2005) uncertainty is 2 Sv.

the overall trends, with one notable exception during the RAPID WB2 mooring failure of late 2005 to early 2006, when the model estimates a stronger southwards UMO than was observed by RAPID.

The model-estimated AMOC transports shown in Figure 8 also agree well with the equivalent estimates from Bryden et al.
285 (2005), with the exception of 1998, where their estimate of 16.1 Sv is just outside the model upper uncertainty of 15.3 Sv. It should be noted however that Bryden et al. (2005) used constant values for both Florida Current and Ekman transports, whereas we used monthly mean values based on observations. The magnitude and trends of the RAPID observations are also captured well by the model estimates from CTD profiles taken during the same period. Including the monthly mean Ekman transport allows the model to capture the 2009–2010 downturn well, although the profiles from November 2009 and October
290 2010 give the only model AMOC transports sufficient weak to be outside the standard deviation of the RAPID monthly mean anomalies of ±2.4 Sv. Prior to 2004, the model estimates the AMOC to be weaker and outside the standard deviation in March of 1987 and 1989, September 1991, June 1993, February 1998, and March 2004, although none reach the magnitude of the 2009–2010 downturn, with the weakest AMOC anomaly of -5.5 Sv seen in March 1987. The strongest model AMOC anomaly of 8.4 Sv is seen on 2 October 2004, which is very close to the RAPID mean anomaly for September 2004 of 8.7 Sv. The
295 AMOC anomaly was also over 2.4 Sv for 11 profiles during the 23 years prior to the start of RAPID and 4 during the 13 years of RAPID observations shown here.

Compared to the LNADW estimates from transatlantic sections made by Bryden et al. (2005), the model predictions get closer with each subsequent section from 1981 to 2004. The model-estimated transports are all more positive than those of Bryden et al. (2005) with the exception of 2004. The LNADW transport estimated by the model for April 2004 differs from
300 that estimated by Bryden et al. (2005) only 0.3 Sv, and from the observed RAPID LNADW transport by less than 0.1 Sv. The model LNADW transports prior to 2004 show a weakening in 1987 to +1 Sv, almost as weak as the observed RAPID monthly

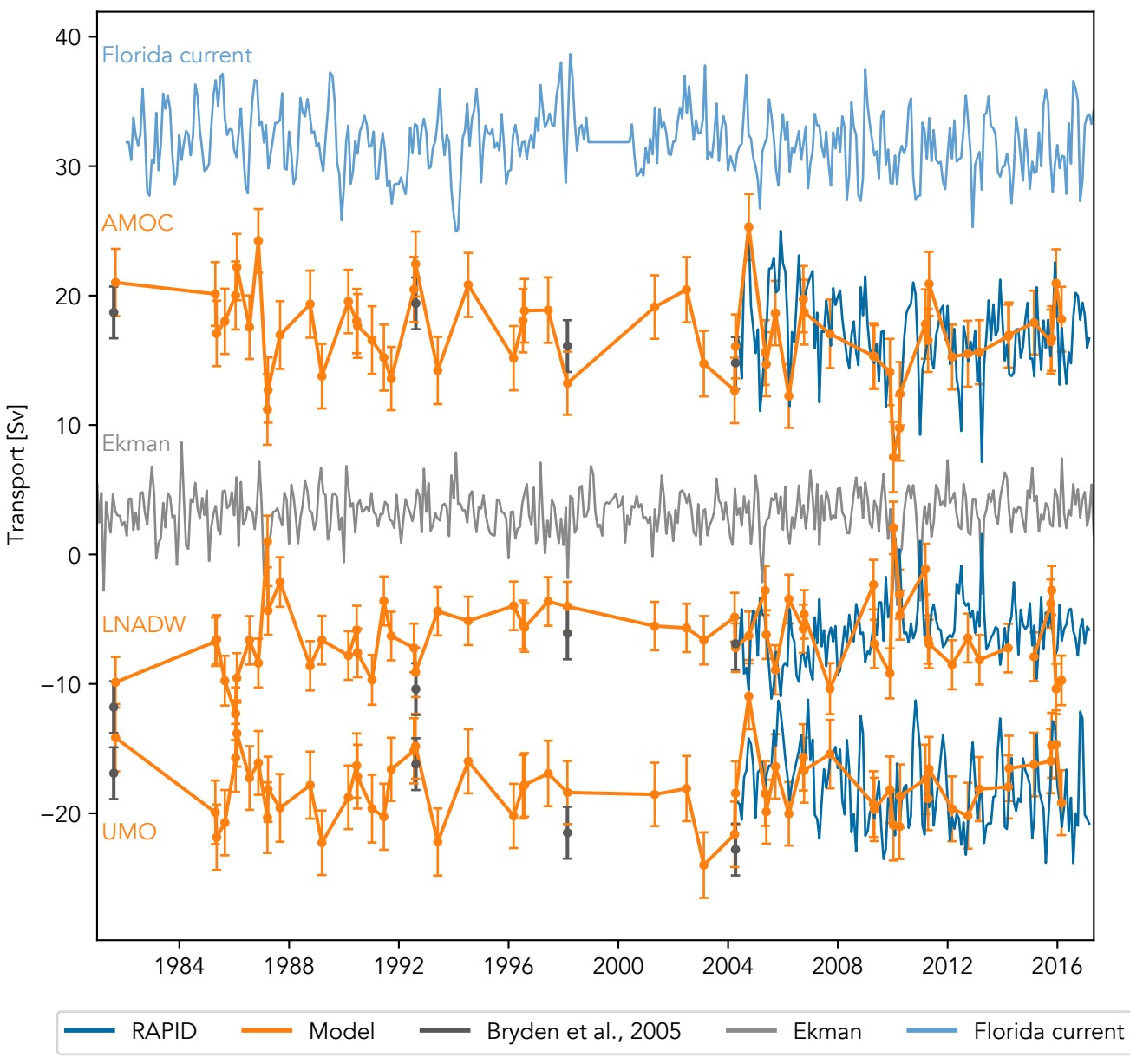

**Figure 8.** UMO and LNADW transports estimated by empirical models using density anomalies from hydrographic CTD profiles, compared to estimates from RAPID and Bryden et al. (2005). The 980 dbar eastern boundary density anomaly for the UMO model was replaced by RAPID monthly climatology. The monthly mean Florida Current and Ekman transports are also shown, and were added to the UMO model-estimated transports to give the estimated AMOC transport.

mean LNADW transports of 1.3, 1.0 and 1.6 Sv in January 2010, December 2010 and March 2013 respectively. The model LNADW transports post-2004 are in reasonable agreement with RAPID trends, although tend to over- or under-estimate the strength. The observed weakening of southwards LNADW flow in 2010 is captured well, with the model LNADW showing a strong positive anomaly.

The four-year mean transports in Figure 9 show that between 1984 and 2000, the UMO strength was within 0.6 Sv of the RAPID mean UMO of -18.3 Sv, taken for the period used to create the model. The four-year means begin from 1984 as prior to this there is only a single profile. The period with the strongest southwards four-year mean UMO is 2000–2004 at -20.6 Sv, lower than the RAPID reduced period of 2008–2012 and 2012–2016 when the mean UMO transports were -18.6 and -18.7 Sv respectively, however the error for the model mean is 4.4 Sv. The four-year model mean UMO transport also compares well to the RAPID equivalent for 2004–2008 and 2008-2012, each differing by only 0.4 Sv. The model mean UMO for 2012–2017 is however 1.6 Sv higher than the RAPID mean. The Gaussian-weighted four-year rolling mean also suggests that the multi-year UMO variability was low during the 1990s, followed by a period of strengthened southward transport in the early 2000s. It also suggest a weakening of southwards UMO transport from 2012 not seen in the observations.

The four-year mean AMOC transports show slightly more variability than the UMO, and agree slightly less well with the RAPID four-year mean values, differing by 0.7 Sv for 2004–2008 and 2008–2012, and 1.5 Sv for 2012–2016. The 2000–2004 mean reflects the UMO downturn, with the lowest four-year mean value of 14.8 Sv, again lower than the 2008–2012 mean AMOC transport for both model and RAPID by 0.4 and 1.1 Sv respectively.

The four-year LNADW rolling mean suggests a non-monotonic weakening trend in the southwards deep return flow between 1985 and 1999, from 8.5 Sv southwards in 1985, decreasing to 3.8 Sv southwards in 1999. The rolling four-year mean then varies by less than 0.6 Sv between 2000 and 2008, then weakens again to between 4.6 and 4.3 Sv southwards in 2009 and 2010 respectively, before increasing in strength again to a maximum southwards transport of 7.7 Sv in 2013. RAPID mean southwards values for 2004–2008 and 2008–2012 are stronger than the model by 0.9 and 0.3 Sv respectively, but for 2012–2016 it is 1.8 Sv weaker. The greater disagreements between model and RAPID mean values for AMOC and LNADW transports may be due to the additional smoothing caused by using monthly mean Ekman transport to estimate them, rather than the 10-day filtered values used by RAPID. None of the model-estimated UMO, AMOC or LNADW transports show an overall trend.

## 4  Discussion

Although the AMOC has been well-observed since 2004 by RAPID, before this, estimates of AMOC transport were restricted to approximately decadal transatlantic sections. It has been estimated that a time series of at least 60 years is necessary to detect long-term change in the AMOC due to anthropogenic global warming (Baehr et al., 2008), so extending the AMOC record into the past is crucial. Although proxies have been used to extend AMOC estimates earlier, these use one, or at most two, layers to represent AMOC dynamics (Longworth et al., 2011; Frajka-Williams, 2015). However, Baehr et al. (2007) showed that deep density measurements were important in reducing the length of the time series required to detect anthropogenic

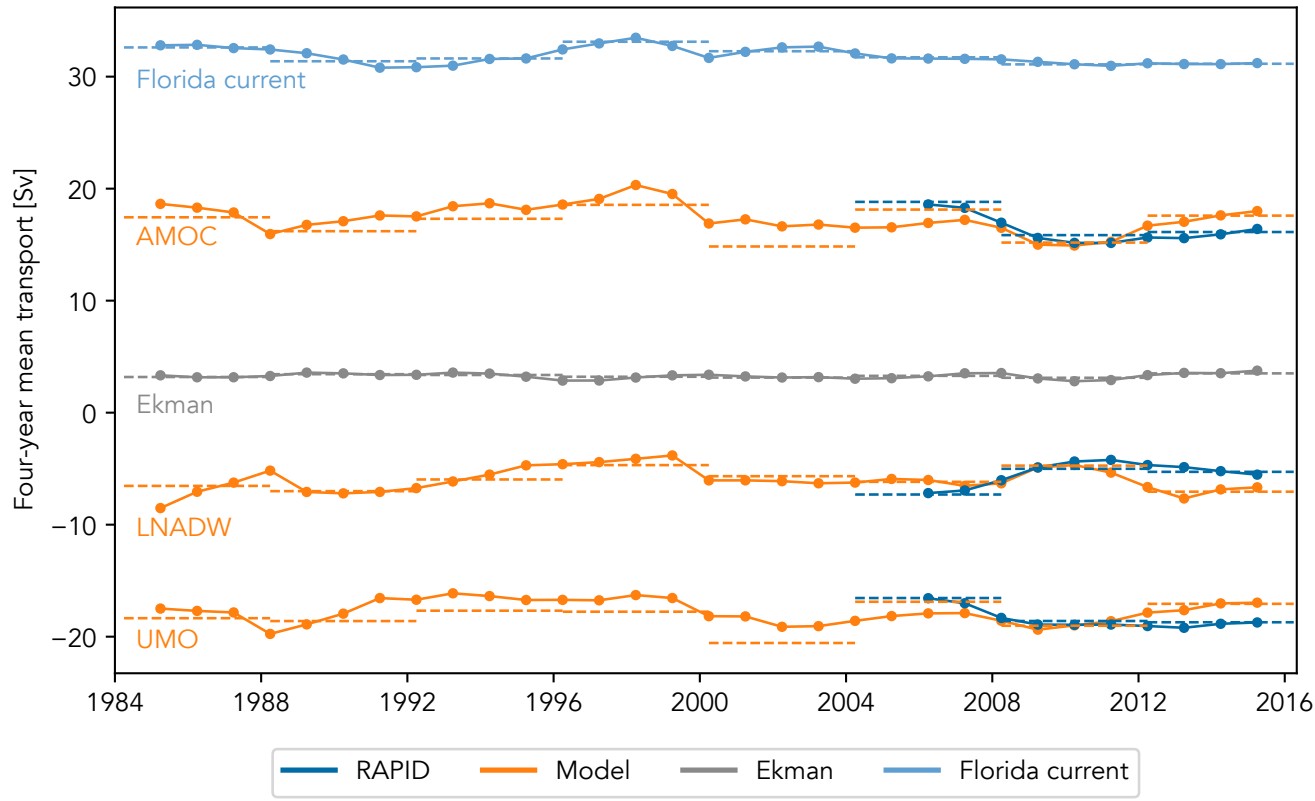

**Figure 9.** Four-year means (dashed lines) from 1984 to 2016 and the Gaussian-weighted rolling mean with a 4-year window (solid line), with the markers showing the mid-point, for AMOC, LNADW and UMO transports estimated by the relevant regression models (orange) and from RAPID observations (dark blue). The four-year and rolling means for Florida Current (light blue) and Ekman (dark grey) transports are also shown.

change, and single layer models neglect this. In this study, we showed that an empirical regression model applied to historical hydrographic data could be used to improve the resolution of the UMO and hence AMOC transport estimates compared to the sparse transatlantic sections. In addition, by representing the deep return layers of the AMOC, the model could capture lower frequency changes missed by other proxy models.

     To develop this empirical model of the AMOC, we regressed UMO transport on western boundary density anomalies within
each of the thermocline, UNADW and LNADW layers, and an eastern boundary climatology within the AAIW layer, using an algorithm to select the best depth for each density anomaly. The selected model was then applied to historical hydrographic CTD profiles to predict UMO and hence AMOC transport strength between 1981 and 2016, at approximately annual resolution. This resolution is sufficient to show pentadal to decadal variability, with a model uncertainty of around ±2.5 Sv.

     There is no overall trend in either AMOC or UMO as estimated by the model, but four-year means, following Smeed et al.
(2018), suggest that there were stronger southwards UMO and stronger northwards AMOC transports between 2000 and 2004

than at any time observed by RAPID. The sea-surface height model developed by Frajka-Williams (2015), which is implicitly single-layer, estimated the mean UMO for 1993–2003 and 2004–2014 to vary by only 0.1 Sv. By contrast, our model estimated the same 11-year means as -19.0 and -18.2 Sv, a difference of 0.8 Sv. The 11-year mean for 1982–1992 was -18.1 Sv, showing that our four-layer model captures more of a change in the UMO. The model mean UMO transport for 2004–2014 of -18.2 Sv

also agrees well with the RAPID equivalent of -17.9 Sv. The importance of representing the deep layers can be seen in repeating the predictions using the two- and three-layer models described earlier. The two-level model representing the upper two layers gives decadal mean differences in UMO transport of 0.3 Sv, while adding the upper deep layer increases the mean UMO for 1982–1992, 1993–2003 and 2004–2014 to -18.3, -18.7 and -18.1 Sv respectively, a difference of 0.6 Sv. The AMOC proxy from Frajka-Williams (2015) had mean transports for 1993–2003 and 2004–2014 of 18.3 and 17.1 Sv, while the RAPID mean

AMOC transport for 2004–2014 was 16.9 Sv. The 11-year mean AMOC transports were predicted by our model as 17.8, 17.4 and 15.9 Sv for 1982–1992, 1993–2003 and 2004–2014 respectively, again showing greater variability than the altimetry-based results.

In addition to the four-layer UMO/AMOC model, we also created a similar model regressing LNADW transport on the deep western boundary density anomaly at 3040 dbar and Ekman transport. The four-year mean LNADW transport estimates from

the same hydrographic profiles show lower frequency variability than the UMO/AMOC, suggesting the deep southwards return flow was strong throughout the late 1980s and 1990s, weakening towards 2000. The four-year mean is also weak during the observed AMOC downturn of 2008–2012. The rolling four-year means for all three transports reflect the changes observed by RAPID well, with decreasing northwards AMOC transport and decreasing deep southwards return flow balanced by an increase in southwards gyre recirculation (Smeed et al., 2018).

Although this model increases the temporal resolution of AMOC estimates, the resolution is still coarse compared to RAPID and the time intervals between profiles are inconsistent. The longest period where no interval is greater than 1 year is October 1988 to July 1994. There are only 2 intervals longer than 2 years: September 1981 to April 1985; and February 1998 to April 2001. The longest interval is 1328 days, the mean is 210 days. Although this resolution is sufficient to show multi-year variability, as shown by the four-year means, the length of some of the sampling intervals and their inconsistency means the

model cannot show interannual variability reliably.

## 5   Conclusions

In conclusion, this study shows that the dynamics of the AMOC can be represented by an empirical linear regression model using boundary density anomalies as proxies for water mass layer transports. More than one layer, represented by boundary density anomalies, are required to capture lower frequency changes to UMO transport. Deep density anomalies combined

with Ekman transport are successful in reconstructing LNADW transport, the deepest limb of the AMOC in the subtropical North Atlantic. Previous proxies for AMOC or UMO at 26°N that rely on single layer dynamics (e.g. Frajka-Williams (2015); Longworth et al. (2011)) cannot capture this low frequency variability. This is also the case for similar reconstructions at other latitudes, for example Willis (2010). Single layer dynamics are also fundamental to estimates of the AMOC that use fixed levels

of no motion such as the MOVE array (Send et al., 2011) or inverted echo sounders (see McCarthy et al. (2020) for details). We have shown the importance of the inclusion of deep density measurements in AMOC reconstructions and believe these to be key to identifying the fingerprint of anthropogenic AMOC change (e.g. Baehr et al. (2008)).

Our model, applied to historical hydrographic data, has increased the resolution of the observed AMOC between 1981 and 2004 from approximately decadal to approximately annual, and in doing so we have shown decadal and four-yearly variability of the AMOC and its associated layer transports. The result is the creation of an AMOC timeseries extending over 3 decades, including for the first time deep density anomalies in an AMOC reconstruction.

Our model has not revealed an AMOC decline indicative of anthropogenic climate change (Stocker et al., 2013) nor the long-term decline reported in SST-based reconstructions of the AMOC (Caesar et al., 2018). It has accurately reproduced the variability observed in the RAPID data, showing that the downturn between 2008 and 2012 (McCarthy et al., 2012) marked not only the weakest AMOC of the RAPID era but the weakest AMOC since the mid-1980s. Since this minimum, the strength of the AMOC has recovered in line with observations from the RAPID array (Moat et al., 2020). In fact, according to our model, southward flowing LNADW has regained a vigour not seen since the 1980s. Recent cold and fresh anomalies in the surface of the North Atlantic subpolar gyre seemed to indicate a return to a cool Atlantic phase associated with a weak AMOC (Frajka-Williams et al., 2017). However, a weakened AMOC was not the primary cause of these anomalies (Josey et al., 2018; Holliday et al., 2020). Whether a restrengthened AMOC will ultimately have a strong impact on Atlantic climate such as was believed to have occurred in the 1990s (Robson et al., 2012) remains to be seen.

*Data availability.* Data from the RAPID AMOC monitoring project are funded by the Natural Environment Research Council and are freely available from http://rapid.ac.uk/rapidmoc/rapid_data/datadl.php (doi: 10.5285/8cd7e7bb-9a20-05d8-e053-6c86abc012c2). The Florida Current cable and section data are made freely available on the Atlantic Oceanographic and Meteorological Laboratory web page (https://www.aoml.noaa.gov/phod/floridacurrent/) and are funded by the DOC-NOAA Climate Program Office - Ocean Observing and Monitoring Division. The World Ocean Database (WOD) is an NCEI product and an IODE (International Oceanographic Data and Information Exchange) project. The work is funded in partnership with the NOAA OAR Ocean Observing and Monitoring Division. WOD2018 data are freely available at https://www.nodc.noaa.gov/OC5/WOD/pr_wod.html. The ECMWF ERA-Interim reanalysis data are freely available at https://apps.ecmwf.int/datasets/.

*Author contributions.* ELW and GDM conceived and designed the study. Analysis was carried out by ELW under supervision by GDM, JVM and RM. BIM and DAS provided software and helped with analysis. ELW prepared the manuscript with contributions from all co-authors.

*Competing interests.* The authors declare that they have no conflict of interest.

*Acknowledgements.* ELW was supported by the Natural Environmental Research Council [grant number NE/L002531/1]. GM was supported by the A4 project (grant aid agreement PBA/CC/18/01) supported by the Irish Marine Institute under the Marine Research Programme funded by the Irish Government, co-financed by the ERDF. This research was supported by EU Horizon 2020 project Blue-Action (grant 727852); and by grants from the UK Natural Environment Research Council for the RAPID-AMOC program and the ACSIS program (NE/N018044/1), by the U.S. National Science Foundation (grant 1332978), by the U.S. National Oceanic and Atmospheric Administration (NOAA) Climate Program Office (100007298), and by the U.S. NOAA Atlantic Oceanographic and Meteorological Laboratory. The authors thank the many officers, crews, and technicians who helped to collect these data.

The authors would like to thank Penny Holliday for her feedback and support of the study, and Eleanor Frajka-Williams for the code used to produce the coherence and phase relationships plots in Figure 5, which is based on the jLab software package (Lilly, 2017).

The authors would also like to thank the two anonymous reviewers whose comments and suggestions helped improve and clarify this manuscript.

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
