# Peer review of "A 30-year reconstruction of the Atlantic meridional overturning circulation shows no decline"

_Ocean Science, 2020_

## Referee Comment (RC1) · Anonymous Referee #1 · 7 Oct 2020

General comments

This manuscript introduces a methodology to increase the AMOC time series beyond the RAPID period. The method uses a linear regression between the density anomaly in the thermocline, intermediate, upper and lower deep layers to compute the AMOC. The strongest result of this study is that the extended time series shows no overall AMOC decline in the period 1981-2016.

Although this result is plausible, my main reservations are related to the method used. Once the authors explain the below questions and introduce the required modifications, the paper is suitable to be published.

Specific comments

[Figure]

Lines 41-57. The authors describe the well known water masses in the North Atlantic Subtropical Gyre (NASG) but several papers should be referenced. For example, Hernández-Guerra et al. (2014) (and references herein) show a carefully description of water masses at 24N that should be mentioned.

Line 42 (Figure 1a). Which year does it correspond the figure to?

Figure 1. I think Figure 1a should show neutral density instead of potential density.

Lines 51-52. It states that AABW flows along the western side of the MAR but Figure 1 suggests that AABW flows in the western and in the eastern side of NASG. The potential density and other properties not shown in this paper do not confirm the presence of such a large amount of AABW at this latitude East of the MAR (and in line 52 it is defined only West of the MAR). I think the plot should be redone.

Lines 52-55. Include References.

Lines 77-78. Zero net flow holds on timescales longer than 10 days was first demonstrated by Kanzow et al (2007).

Line 87. Explain the selection of 4820 dbar as reference level, related to the change from northward AABW to southward LNADW. Include references previous to McCarthy (2015), (Bryden 2009, Kanzow 2007).

Lines 92-93. Smeed et al 2014 stated that "the majority of the change in the AMOC is associated with the UMO transport". Therefore, the UMO is the main contributor to AMOC changes, as the main contributor to the AMOC is the Florida Strait transport (with higher net values). Repeated in lines 120-122.

Section 2.2. I have a very strong concern about the model. Figure 2 plots the thermocline transport anomaly on the temperature anomaly at 400 dbar and a linear regression adjusting the data. What I can see in Figure 2 is a strong scattering of points that any linear regression could adjust as the authors find (only 20% of the variance is adjusted). From here, authors try to find another depth which could explain a higher

variance. They find that at 780 dbar, the explained variance increases to 51% but any figure is shown with the data and the linear regression.

Lines 125-128. After Chidichimo et al. (2010), several papers have appeared dealing with the seasonal cycle of the AMOC and the eastern boundary of the NASG. Pérez-Hernández et al.(2015), Vélez-Belchí et al. (2017), Hernández-Guerra et al. (2017) and Casanova-Masjoan et al. (2020), among others, have found a seasonal behaviour of the Canary Current in the Lanzarote Passage that explains the seasonal cycle of the AMOC. I think these papers deserve at least a brief comment in the manuscript.

Lines 128-131. The method uses four variables to relate the AMOC and density anomalies at different depths. The use of the density anomalies related to the intermediate layer significantly increases the R2 from 0.49 to 0.74. In contrast, the deep density anomalies (UNADW and LNADW) only account for 2% of R2. This 2% explained by these two variables could be below the noise of each variable. I suggest to carry out a Monte Carlo method to estimate an uncertainty and to check that this 2% is above the statistical uncertainty.

Line 149 (Figure 3). Include uncertainties of RAPID measurements.

Lines 166-167. I am wondering if the western boundary density anomaly could also be replaced with monthly climatology as in the eastern boundary.

Line 174. That is not the typical definition of the standard deviation.

Lines 188-191. Would it be possible to assess the cross correlation with a wavelet transform to add information to the coherence? That way we could assess the change in power spectra for each frequency through the years. If authors think that this study is going to take too long, please, do not do it. It is only a suggestion.

Figure 5 (right panels). The phase or lag in degrees does not provide very useful information. It could be expressed in time (days), so that we could estimate the time lag between each signal. Moreover, the dashed line of "out of phase" at 180° may induce

[Figure]

to errors in the reader, as any signals are out of phase once the phase is different from zero.

Line 193. I think it should be written: '... shows significant coherence with the observed UMO transport at periods of ...'

Line 197. What is the consistency between a 180° coherence phase and a negative coefficient in the model regression? Relate to the peak/valleys in each signal

Line 200. Please, explain the negative phase. Previously, signal A was anticipated to signal B (positive phase), while now signal A is delayed with respect to signal B (negative phase). Make sure to define which are signals A and B, so that we may understand if the UMO transport or the boundary densities are lagged with respect to each other.

Lines 203-204. What is the relation between a 90° phase coherence and a weakening of the southward UMO transport?

Line 229. "Losing a little of the explained UMO transport variance". Would there be any way to assess the contribution of the seasonal component to the UMO variability?

Minor comments:

Line 23. (IPCC) says

Line 154. Change doesn't to does not.

Line 183. The Florida Current data have a gap.

Lines 212-214. Missing the verb in the sentence.

Line 241. If there are 5 CTD profiles within the group, with distances from the slope at 740 dbar of

Lines 244-245. those selected for use in the model, show that the majority were

Line 247. Transatlantic sections at 24.5°N (consistent with previous sections).

Line 250. The uncertainty for the 1957 section model estimate was much larger

Lines 252-255. Could be separated into two sentences.

Line 273. Badly worded.

Line 284. Almost as weak as

Line 285: December 2010 and March 2013, respectively.

Line 295: The Gaussian-weighted four-year rolling mean also suggests that

Line 297: It also suggests

Line 302: suggests a non-monotonic weakening trend

Line 305: maximum

Lines 306-307: RAPID mean southward values for 2004-2008 and 2008-2012 are stronger than the model by 0.9 and 0.3 Sv respectively, but for 2012-2016 it is 1.8 Sv weaker.

Lines 320-321: How low are the targeted low frequencies? The model time series only allows for decadal changes ($\sim$30 years of model).

Line 326: Using four-year means, how can you observe multi-year variability?

Lines 175-179, 206, …. I am not sure why the authors start to use the past tense (calculated, used, suggested, …)

References

Casanova-Masjoan, M., Pérez-Hernández, M.D., Vélez-Belchí, P., Cana, A., Hernández-Guerra, A., 2020. Variability of the Canary Current diagnosed by inverse box models. Journal of Geophysical Research - Oceans, 125, e2020JC016199, https://doi.org/10.1029/2020JC016199

Hernández-Guerra, A., Espino-Falcón, E., Vélez-Belchí, P., Pérez-Hernández, M. D.,

Martínez-Marrero, A., Cana, L., 2017. Recirculation of the Canary Current in fall 2014. Journal of Marine Systems, 174, 25-39. https://doi.org/10.1016/j.jmarsys.2017.04.002

Hernández-Guerra, A., Pelegrí., J. L., Fraile-Nuez, E., Benítez-Barrios, V. M., Emelianov, M., Pérez-Hernández, M. D., Vélez-Belchí, P., 2014. Meridional overturning transports at 7.5N and 24.5N in the Atlantic Ocean during 1992–93 and 2010–11. Progress in Oceanography, 128, 98-114. http://dx.doi.org/10.1016/j.pocean.2014.08.016

Pérez-Hernández, M. D., McCarthy, G. D., Velez-Belchí, P., Smeed, D. A., Fraile-Nuez, E., Hernández-Guerra, A., 2015. The Canary Basin contribution to the seasonal cycle of the Atlantic Meridional Overturning Circulation at 26° N. Journal of Geophysical Research: Oceans, 120(11), 7237-7252. http://dx.doi.org/10.1002/2015JC010969

Vélez-Belchí, P., Pérez-Hernández, M. D., Casanova-Masjoan, M., Cana, L., Hernández-Guerra, A., 2017. On the seasonal variability of the Canary Current and the Atlantic meridional overturning circulation. Journal of Geophysical Research, 122(6), 4518-4538. http://dx.doi.org/10.1002/2017JC012774

---

## Referee Comment (RC2) · Anonymous Referee #2 · 3 Nov 2020

Manuscript #: os-2020-71

This study presents a new linear regression model to estimate the AMOC strength at 26degN and its two subcomponents, i.e. the upper mid-ocean transport and the Lower North Atlantic Deep Water transport, back to 1981 based on the density anomalies in the western boundary. In particular, the new approach allows the temporal resolution of the time series to be nearly annual, thus sufficiently resolving interannual variability on timescale of ∼4 years. The main conclusion is that this new AMOC time series does not exhibit any significant weakening trend throughout the record. This is an excellent study with a clever method and clear presentation. I only have some minor comments as listed below.

1. L108-110, Figure 2: The rather nontrivial difference between the calculation in this

[Figure]

paper and that by Longworth et al. (2011) may suggest a nonstationary relationship between the ∼400 db temperature and the thermocline transport. Such aspect may have an implication for the reconstruction method employed by the authors, as the multiple regression is trained for the RAPID period and applied to a much longer period. Therefore, the cross-validation approach for training the multiple regression would allow a quantification of uncertainty due to potential nonstationarity. The training period can be broken into ∼3 segments and the regression coefficients for each segment can be measured by fitting the model to the rest of the time series. Then, the end results such as the Fig. 3 can be constructed by stitching the regressions from all the segments together. The authors tried a cross-checking by testing the model on the latest 21 month RAPID data that were not used in the model training (L163-166). However, a systematic cross-validation would allow a more robust estimate of the uncertainty.

2. L146: What is the interpretation for the autocorrelation being significant for a lag of one month?

3. L200: "(Figure 5e), For" <- The comma should be a period.

4. L231: "during Longworth et al." -> "by Longworth et al."

5. L466: Please correct some of the broken symbols.

---

## Author Comment (AC1) · 8 Dec 2020

General comments

This manuscript introduces a methodology to increase the AMOC time series beyond the RAPID period. The method uses a linear regression between the density anomaly in the thermocline, intermediate, upper and lower deep layers to compute the AMOC. The strongest result of this study is that the extended time series shows no overall AMOC decline in the period 1981-2016. Although this result is plausible, my main reservations are related to the method used. Once the authors explain the below ques-

[Figure]

tions and introduce the required modifications, the paper is suitable to be published.

Specific comments

Lines 41-57. The authors describe the well known water masses in the North Atlantic Subtropical Gyre (NASG) but several papers should be referenced. For example, Hernández-Guerra et al. (2014) (and references herein) show a careful description of water masses at 24N that should be mentioned.

Thank you for this feedback, lines 41-57 will be re-written to improve the description of the water masses and reference Hernández-Guerra et al. (2014), Fraile-Nuez et al., 2010, Hernández-Guerra et al., 2003, Machín et al., 2006, Talley & McCartney 1982, and Pickart 1992.

Line 42 (Figure 1a). Which year does it correspond the figure to?

We will add the relevant citation from 2011 for the WOCE Ocean Atlas section.

Figure 1. I think Figure 1a should show neutral density instead of potential density.

Figure 1a will be changed to show the WOCE Atlas section of neutral density at 24°N.

Lines 51-52. It states that AABW flows along the western side of the MAR but Figure 1 suggests that AABW flows in the western and in the eastern side of NASG. The potential density and other properties not shown in this paper do not confirm the presence of such a large amount of AABW at this latitude East of the MAR (and in line 52 it is defined only West of the MAR). I think the plot should be redone.

Thank you for catching this error. The schematic will be altered to show AABW only on the western side of the MAR, and LNADW filling the deepest part of the eastern basin.

Lines 52-55. Include References.

Please see our response to comment #1.

Lines 77-78. Zero net flow holds on timescales longer than 10 days was first demonstrated by Kanzow et al (2007).

Thank you, this citation will be added.

Line 87. Explain the selection of 4820 dbar as reference level, related to the change from northward AABW to southward LNADW. Include references previous to McCarthy (2015), (Bryden 2009, Kanzow 2007).

A sentence explaining the reference level and referencing McCarthy et al., 2015 will be added. We are only explaining the current RAPID methodology, so have not included the earlier references in the interest of brevity.

Lines 92-93. Smeed et al 2014 stated that "the majority of the change in the AMOC is associated with the UMO transport". Therefore, the UMO is the main contributor to AMOC changes, as the main contributor to the AMOC is the Florida Strait transport (with higher net values). Repeated in lines 120-122.

Thank you for noticing this repetition, the first instance (line 93) will be removed.

Section 2.2. I have a very strong concern about the model. Figure 2 plots the thermocline transport anomaly on the temperature anomaly at 400 dbar and a linear regression adjusting the data. What I can see in Figure 2 is a strong scattering of points that any linear regression could adjust as the authors find (only 20% of the variance is adjusted). From here, authors try to find another depth which could explain a higher variance. They find that at 780 dbar, the explained variance increases to 51% but any figure is shown with the data and the linear regression.

In response to the reviewer's suggestion to include an additional figure, we will add Figure 2b, to show the higher variance explained by the different depth. But the fundamental point of this one layer model not being a good model is one that we concur with. We take the low explained variance as motivation for developing more sophisticated models within this paper.

Lines 125-128. After Chidichimo et al. (2010), several papers have appeared dealing

with the seasonal cycle of the AMOC and the eastern boundary of the NASG. Pérez-Hernández et al.(2015), Vélez-Belchí et al. (2017), Hernández-Guerra et al. (2017) and Casanova-Masjoan et al. (2020), among others, have found a seasonal behaviour of the Canary Current in the Lanzarote Passage that explains the seasonal cycle of the AMOC. I think these papers deserve at least a brief comment in the manuscript.

Thank you for the suggestion, this section will be changed to read:

The seasonal cycle of the AMOC is driven largely by seasonality at the eastern boundary (Chidichimo et al., 2010; Pérez-Hernández et al., 2015). The annual maximum northwards transport at the eastern boundary and the AMOC occur around October (Vélez-Belchí et al., 2017), and is driven by changes in the circulation of the Canary Current (Casanova-Masjoan et al., 2020; Hernández-Guerra et al., 2017), and at intermediate depths ( 700-1400 dbar) by seasonal changes in the Intermediate Poleward Undercurrent (Hernández-Guerra et al., 2017; Vélez-Belchí et al., 2017). Eastern boundary density anomalies have maximum sub-surface variability around 1000 dbar (Chidichimo et al., 2010), so the AAIW layer was represented by an eastern boundary density anomaly between 800 and 1100 dbar.

Lines 128-131. The method uses four variables to relate the AMOC and density anomalies at different depths. The use of the density anomalies related to the intermediate layer significantly increases the R2 from 0.49 to 0.74. In contrast, the deep density anomalies (UNADW and LNADW) only account for 2% of R2. This 2% explained by these two variables could be below the noise of each variable. I suggest to carry out a Monte Carlo method to estimate an uncertainty and to check that this 2% is above the statistical uncertainty.

We have taken the reviewer's suggestion to assess the significance of the model improvements. We have undertaken a Monte Carlo simulation by assessessing the effects of random noise on the model to assess how much of the model improvement due to inclusion of the deeper layers is chance. The mean adjusted R-squared value

is 0.73, with a standard deviation of 0.004. We conclude that the improvement is not simply due to chance. Additionally, the deep density anomalies are both significant ($p < 0.05$) and independent (show no collinearity). However, our arguments for including the deeper layers are not simply statistical. The AMOC is a three-dimensional phenomenon and including the deeper layers makes sense (cf. Figure 1). The importance of including the deeper layers is further emphasized by our analysis of the timescales of variability (Fig. 5). The deeper layers are more important at longer timescales—as would be expected (e.g. Moat et al. 2020), and as lines 333-340 describe, the reduction in decadal mean UMO and AMOC expected due to the reduced AMOC observed by RAPID is only seen when all four variables are included.. Therefore we argue that their inclusion in the model is both statistically and dynamically significant.

Line 149 (Figure 3). Include uncertainties of RAPID measurements.

The uncertainty of 1.5 Sv determined by McCarthy et al., 2015 for 10-day filtered RAPID transports will be added to Figure 3.

Lines 166-167. I am wondering if the western boundary density anomaly could also be replaced with monthly climatology as in the eastern boundary.

The only reason for using the eastern boundary climatology was to allow the use of western boundary CTD profiles without corresponding eastern profiles, and we make use of the strong seasonality at the eastern boundary to do so. We allow that using the western boundary density anomaly to create a regression model is feasible, but are not sure what advantage that would bring, given that it would just create a climatology model.

Line 174. That is not the typical definition of the standard deviation.

This was poorly worded, and should state that the error was taken to be the difference between the predicted and observed UMO, and that the standard deviation of the bootstrapped errors was 2.8 Sv. The text will be changed to make this clearer.

Lines 188-191. Would it be possible to assess the cross correlation with a wavelet transform to add information to the coherence? That way we could assess the change in power spectra for each frequency through the years. If authors think that this study is going to take too long, please, do not do it. It is only a suggestion.

Thank you, we believe that it is outside the scope of this study, but it is an excellent suggestion that we will consider for future analysis of these results.

Figure 5 (right panels). The phase or lag in degrees does not provide very useful information. It could be expressed in time (days), so that we could estimate the time lag between each signal. Moreover, the dashed line of "out of phase" at 180◦ may induce to errors in the reader, as any signals are out of phase once the phase is different from zero.

We believe that the phrase 'out of phase' is generally understood to mean a phase of $180°$, so the peak and trough of the two signals co-vary. We will add a sentence to the caption of Figure 5 to make this clear. The phase in time will depend on the period of each signal, which is different for each point on the x-axis, so the phase must be expressed in degrees.

Line 193. I think it should be written: '. . . shows significant coherence with the observed UMO transport at periods of . . .'

Thank you for the feedback, the sentence will be rewritten as suggested.

Line 197. What is the consistency between a 180◦ coherence phase and a negative coefficient in the model regression? Relate to the peak/valleys in each signal

The eastern boundary density anomaly has a negative coefficient in the model regression, so a decrease in density at the eastern boundary is associated with an increase in northwards UMO transport. Figure 5d shows that a peak in the UMO signal is associated with a trough in the eastern boundary density anomaly signal. Figure 5c shows at what periods this co-variability is significant.

[Figure]

Line 200. Please, explain the negative phase. Previously, signal A was anticipated to signal B (positive phase), while now signal A is delayed with respect to signal B (negative phase). Make sure to define which are signals A and B, so that we may understand if the UMO transport or the boundary densities are lagged with respect to each other.

The text will be changed to state whether the boundary density anomaly is lagged with respect to the UMO or vice versa.

Lines 203-204. What is the relation between a 90âŬę phase coherence and a weakening of the southward UMO transport?

This statement will be removed.

Line 229. "Losing a little of the explained UMO transport variance". Would there be any way to assess the contribution of the seasonal component to the UMO variability?

The contribution of the seasonal component and the loss of the explained UMO transport variance can be seen in Figure 4a, where we compare the predicted UMO using the eastern boundary density anomaly and an eastern boundary climatology.

Minor comments:

Line 23. (IPCC) says

'say' will be changed to 'says'

Line 154. Change doesn't to does not.

'doesn't' will be changed to 'does not'

Line 183. The Florida Current data have a gap.

'has' will be changed to 'have'

Lines 212-214. Missing the verb in the sentence.

Thank you, the verb will been added and the sentence re-written to improve clarity.

Line 241. If there are 5 CTD profiles within the group, with distances from the slope at 740 dbar of

'720 dbar' will be corrected to '740 dbar'

Lines 244-245. those selected for use in the model, show that the majority were

'shows' will be changed to 'show'

Line 247. Transatlantic sections at 24.5◦N (consistent with previous sections).

'25' will be changed to '24.5'

Line 250. The uncertainty for the 1957 section model estimate was much larger

'were' will be changed to 'was'

Lines 252-255. Could be separated into two sentences.

Thank you, this will be altered as suggested.

Line 273. Badly worded.

This sentence will be split into two rewritten sentences to improve clarity.

Line 284. Almost as weak as

'almost weak as' will be changed to 'almost as weak as'

Line 285: December 2010 and March 2013, respectively.

'December 2010 March 2013' will be changed to 'December 2010 and March 2013'

Line 295: The Gaussian-weighted four-year rolling mean also suggests that Line 297: It also suggests

'also suggest that' will be changed to 'also suggests that'

Line 302: suggests a non-monotonic weakening trend

Thank you, this will be changed.

Line 305: maximum

'maximn' will be corrected to 'maximum'

Lines 306-307: RAPID mean southward values for 2004-2008 and 2008-2012 are stronger than the model by 0.9 and 0.3 Sv respectively, but for 2012-2016 it is 1.8 Sv weaker.

This will be changed to the suggested sentence, thank you.

Lines 320-321: How low are the targeted low frequencies? The model time series only allows for decadal changes (âĹij30 years of model).

This is covered in the change to line 326.

Line 326: Using four-year means, how can you observe multi-year variability?

'Multi-year' will be changed to pentadal to avoid confusion.

Lines 175-179, 206, .... I am not sure why the authors start to use the past tense (calculated, used, suggested, . . .)

We believe the past tense is used consistently except when describing figures, or facts that remain true (e.g., the R-squared value of a linear regression).

References Casanova-Masjoan, M., Pérez-Hernández, M.D., Vélez-Belchí, P., Cana, A., Hernández-Guerra, A., 2020. Variability of the Canary Current diagnosed by inverse box models. Journal of Geophysical Research - Oceans, 125, e2020JC016199, https://doi.org/10.1029/2020JC016199 Hernández-Guerra, A., Espino-Falcón, E., Vélez-Belchí, P., Pérez-Hernández, M. D., Martínez-Marrero, A., Cana, L., 2017. Recirculation of the Canary Current in fall 2014. Journal of Marine Systems, 174, 25-39. https://doi.org/10.1016/j.jmarsys.2017.04.002 Hernández-Guerra, A., Pelegrí.,

J. L., Fraile-Nuez, E., Benítez-Barrios, V. M., Emelianov, M., Pérez-Hernández, M. D., Vélez-Belchí, P., 2014. Meridional overturning transports at 7.5N and 24.5N in the Atlantic Ocean dur- ing 1992–93 and 2010–11. Progress in Oceanography, 128, 98-114. http://dx.doi.org/10.1016/j.pocean.2014.08.016 Pérez-Hernández, M. D., McCarthy, G. D., Velez-Belchí, P., Smeed, D. A., Fraile-Nuez, E., Hernández-Guerra, A., 2015. The Canary Basin contribution to the seasonal cycle of the Atlantic Meridional Overturning Circulation at 26°e N. Journal of Geophysical Research: Oceans, 120(11), 7237-7252. http://dx.doi.org/10.1002/2015JC010969 Vélez-Belchí, P., Pérez-Hernández, M. D., Casanova-Masjoan, M., Cana, L., Hernández-Guerra, A., 2017. On the seasonal variability of the Canary Current and the Atlantic meridional overturning circulation. Journal of Geophysical Research, 122(6), 4518-4538. http://dx.doi.org/10.1002/2017JC012774

---

## Author Comment (AC2) · 8 Dec 2020

This study presents a new linear regression model to estimate the AMOC strength at 26degN and its two subcomponents, i.e. the upper mid-ocean transport and the Lower North Atlantic Deep Water transport, back to 1981 based on the density anomalies in the western boundary. In particular, the new approach allows the temporal resolution of the time series to be nearly annual, thus sufficiently resolving interannual variability on timescale of âĹij4 years. The main conclusion is that this new AMOC time series does

not exhibit any significant weakening trend throughout the record. This is an excellent study with a clever method and clear presentation. I only have some minor comments as listed below.

1. L108-110, Figure 2: The rather nontrivial difference between the calculation in this paper and that by Longworth et al. (2011) may suggest a nonstationary relationship between the âĹij400 db temperature and the thermocline transport. Such aspect may have an implication for the reconstruction method employed by the authors, as the multiple regression is trained for the RAPID period and applied to a much longer period. Therefore, the cross-validation approach for training the multiple regression would allow a quantification of uncertainty due to potential nonstationarity. The training period can be broken into âĹij3 segments and the regression coefficients for each segment can be measured by fitting the model to the rest of the time series. Then, the end results such as the Fig. 3 can be constructed by stitching the regressions from all the segments together. The authors tried a cross-checking by testing the model on the latest 21 month RAPID data that were not used in the model training (L163-166). However, a systematic cross-validation would allow a more robust estimate of the uncertainty.

The reviewer is correct in that although the 400 dbar temperature and thermocline transport anomalies exhibit stationary behaviour themselves, the residuals of the regression on them are non-stationary (with stationarity determined using the Augmented Dickey-Fuller. However, the multiple linear regression selected has stationary residuals, although two out of the four exogenous variables are non-stationary.

Following the reviewer's suggestion, we have repeated the cross validation by using the first and last 30% of the original training data to train the model, and the remaining 70% as test data. Although the coefficients differ, each 30%-model gives a Pearson's correlation coefficient of 0.88 between the model-predicted UMO and the observed UMO, higher than the 'full' model (Figures 1 and 2).

Also, used to predict the UMO for the latest 21 month RAPID data, the reduced time

series models give Pearson's correlation coefficients of 0.75 and 0.73 for the models trained on the first and last 30% of the original training data respectively. This compares to r=0.75 for the full model.

Finally, the predictions were also validated against hydrographic data from the period of the RAPID project, and against transatlantic section data from 1981 onward. These results lead us to believe that the multiple linear regression model we develop in this paper is valid for predicting longer time series, and has a realistic estimate of the uncertainty. We will add a brief description of the cross-validation to the manuscript.

2. L146: What is the interpretation for the autocorrelation being significant for a lag of one month?

As autocorrelation is inherent in regression models based on time series, and the RAPID data shows strong seasonality, we would expect to see a relationship between monthly averaged data, in this case between one month and the next but not significantly beyond that. This result is consistent with Smeed et al., 2014, who show the decorrelation length scale associated with the AMOC is 40 days. This last reference and comment will be added to the manuscript.

3. L200: "(Figure 5e), For" <- The comma should be a period.

This will be corrected in the manuscript.

4. L231: "during Longworth et al." -> "by Longworth et al."

This will be corrected in the manuscript.

5. L466: Please correct some of the broken symbols.

These will be corrected in the manuscript.
* * *
[Figure]

**Fig. 1.** UMO transport anomaly predicted from last 70% of training data, using model created from first 30% of training data

[Figure]

**Fig. 2.** UMO transport anomaly predicted from first 70% of training data, using model created from last 30% of training data